# Convolutional Autoencoding of Small Targets in the Littoral Sonar Acoustic Backscattering Domain

Timothy J. Linhardt [1,*] , Ananya Sen Gupta [1] and Matthew Bays [2]

1   Department of Electrical and Computer Engineering, University of Iowa, Iowa City, IA 52242, USA
2   Panama City Division, Naval Surface Warfare Center, Panama City, FL 32407, USA
*   Correspondence: timothy-linhardt@uiowa.edu

**Abstract:** Automated target recognition is an important task in the littoral warfare domain, as distinguishing mundane objects from mines can be a matter of life and death. This is initial work towards the application of convolutional autoencoding to the littoral sonar space, with goals of disentangling the reflection noise prevalent in underwater acoustics and allowing recognition of the shape and material of targets. The autoencoders were trained on magnitude Fourier transforms of the TREX13 dataset. Clusters in the encoding space representing the known variable of measurement distance between the target and the sensor were found. An encoding vector space of around 16 dimensions appeared sufficient, and the space was shown to generalize well to unseen data.

**Keywords:** acoustics; underwater; autoencoder; convolution; encoding; littoral; machine learning; object detection; feature representation

## 1. Introduction

The oceanic acoustic environment poses many challenges to autonomous target characterization not present in other signaling domains. Reflections off the surface of the ocean and the seafloor, as well as clutter like fish, rocks and bubbles add noise and echoes that greatly impede clarity [1]. Robustly defining and selecting features for underwater signals is necessary for automatic target recognition tasks [2–6].

We want to apply techniques from the deep learning community [7] and provide new methods of target representation and analysis, specifically employing the various types of autoencoders. Deep learning has had recent successes in the underwater sonar domain such as in [8] where it was used to automate sonar processing.

Our end goal is to develop a model that can robustly provide insight into acoustic backscattering returns for object detection with application to mine detection [9]. This would at minimum constitute object detection and ordinance classification regardless of noise, to be utilized in conjunction with an autonomous underwater vehicle [10,11].

The goal of informing object classification, also called automated target recognition (ATR), via geometric characteristics detected in the acoustic signal is of particular interest for maritime search [12–15]. Unlike other applications such as natural language processing, the nature of the maritime environment causes the cost of data collection for ATR training purposes to be extraordinarily high, involving days of on-water tests to collect a limited set of sonar returns on relatively similar targets within a target field [16]. The high cost of data collection necessitates other methods for data collection, including the development of simulated data [17], aggregation of heterogeneous datasets [18], and physics-based parameter characterisation during the ATR process [19,20]. Among these data-centric approaches, physics-based methods show considerable promise due to the direct involvement of the model in the learning process.

In maritime applications, it is often the case that while acoustic imagery is unavailable, engineering parameters related to a target's shape, size, and other characteristics are known.

The interplay between these physical traits and the sonar returns allows for the potential to train algorithms to detect these physical features with increased accuracy for a given amount of training data compared to other methods. This is due to the direct involvement of the physical characteristics without secondary approximations that may occur when training on simulated data.

The key concept to be applied to the acoustic backscattering data is the autoencoder. One of the earliest uses of an autoencoder was [21] in 1987 as documented in [22]. Many variants and extensions of this concept have developed since then, but the core idea remains. A network is trained in an unsupervised manner such that inputs of high dimension are reduced to lower dimensional encodings akin to the classic principal component analysis (PCA) technique.

This key dimensional advantage provides the background motivation for adopting the autoencoder for compact yet effective feature representation of sonar backscattering from small targets such as considered in this work. Lower-dimensional feature engineering are inherently less vulnerable to unstructured noise from the sonar environment, and also provide a computational advantage towards eventual target classification. The backscattering data is a two dimensional signal domain, so our autoencoders are convolutional by construction, with an aim to provide smoothly connected lower-dimensional feature encoding of the targets of interest. The main disadvantage of an autoencoder is that it is a physics-agnostic and purely computational tool. Therefore, it is unable to provide direct physical interpretation of the individual values in the encodings that the network produces.

The scope of this work is constrained to explore the representational power of the autoencoder in the context of real sonar targets over field data. In this context, given the noise suppression and computational advantages inherent in lower-dimensional feature encoding, this lack of immediate physical interpretation does not pose a fundamental limitation towards successful feature encoding and subsequent reconstruction. Therefore, the compact representational advantage makes the autoencoder a useful tool towards lower-dimensional feature engineering, and in future work we aim to explore the physical meaning of autoencoded features using extensive simulations of a variety of sonar targets under diverse environmental conditions.

The key contributions of this work are as follows.

(i)     We take the preexisting convolutional autoencoder network concept [23] and apply it to a different data domain: underwater sonar backscattering from small elastic targets.
(ii)    We approximate the dimensionality of a feature space for this new data through empirical analysis, and then
(iii)   visualize the multimodal network reconstruction error distributions collected through 5-fold cross-validation.

This enables feature interpretation based on the autoencoder framework in terms of potentially lower dimensional representations.

In future work, we will aim for encoding vector interpretability, similar to how the InfoGAN [24] works. InfoGAN is a type of generative adversarial network (GAN), similar in concept to an inverted autoencoder [25,26], that generates data from an interpretable vector of variables with defined units, and a noise vector. Out goal is disentangling a noise vector from an interpretable coding vector that would contain object class as well as useful information that could include distance from the target, measurement angle, and ground sediment type. Additionally, we may branch off to use such results to generate underwater acoustic signal in a different way from existing techniques such as in [27].

## 2. Materials and Methods

### 2.1. Principle Component Analysis

Autoencoders take inspiration from the process of principal component analysis (PCA) [28], a method by which data dimensionality is reduced. Given a set of data vectors, one finds a a linear projection to a lower dimensional space, or principle subspace, while



maximizing the variance of the projected data. This is useful for data compression and feature extraction [29,30].

Given a distribution of points in a high dimensional vector space, the eigenvectors of the covariance matrix form a new set of perpendicular components, with the highest magnitude eigenvalue associated with the most principle component, and in sequence of magnitudes beyond that. To perform dimensionality reduction, the component space is projected onto a lower dimension principle subspace that is created by removing the least principle components.

### 2.2. Autoencoders

Autoencoding networks learn a low dimensional subspace to the space of the input data. Unlike with PCA, the components of the encoding are not orthogonal and linear vectors, but rather nonlinear functions on an input and the network's trained weights [30]. The goal is not perfect reconstruction; instead important characteristics are ideally emphasized by reducing the inputs to a tight bottleneck [23].

An autoencoder is a neural network comprised of two subnetworks in sequence. The first subnetwork is called the encoder, and performs dimension reduction, resulting in the encoding. This encoding is then fed into the second subnetwork, referred to as a decoder, and it attempts to reverse the effects of the encoder.

The encoder learns a mapping from the input data space to a low dimension vector space, Equation (1), which is the encoding. In our use case, the input space is a set of two dimensional signals that can be represented as $M \times N$ matrices of real values. The matrices are fed through a series of decimating convolutional layers separated with regularization and nonlinear activation layers. The result of the final convolution has a regularization and nonlinear activation applied to it before it is reshaped into a vector. This vector has a dense neuron layer applied to it to produce the encoding.

$$E : \mathbb{R}^{M \times N} \to \mathbb{R}^k \tag{1}$$

The decoder attempts to reverse the mapping performed by the encoder, Equation (2), with a sequence of dilating transposed convolutional layers to reconstruct the input data. The structure is effectively an inverse of the encoder's: a dense layer followed by regularization and activation and then the transposed convolutional layers with the same separation layers. The final transposed convolutional layer produces the reconstruction attempt.

$$D : \mathbb{R}^k \to \mathbb{R}^{M \times N} \tag{2}$$

It is in practice impossible to have $D = E^{-1}$ because of the lossy mapping, so this process finds the most important features of the input space.

PCA performed on input images yields component images that can be linearly combined while decoding, but the components of autoencoders are not orthogonal, so this cannot be mimicked.

The training process of autoencoders is the same as with other neural networks. Backpropagation and stochastic gradient descent are utilized to minimize a loss function. Batch normalization layers are utilized to reduce generalization error [31] and constrain the values between the network layers [32]—to reduce the likelihood of exploding gradients.

The loss function for an autoencoder is related to how the error between the input and output is evaluated. The sum of the element-wise mean squared error (MSE) between the input and reconstruction is a common choice for the error metric in autoencoding.

$$\text{MSE}(x, D(E(x))) = \sum_{i=1}^{M} \sum_{j=1}^{N} x_{ij} - D(E(x))_{ij} \tag{3}$$

Signal-like data is typically passed through convolutional neuron layers in neural networks instead of the more general-case fully connected layers that are tantamount to matrix multiplication [33].

The autoencoder can be called "sparse" if a sparsity constraint is applied to the values of the encoding vector [34]. Sparse autoencoders are used to find interpretable features for in other tasks [23]. It penalizes the loss function if the expected value an individual component of the encoding diverges from a small value, with the expected value estimated for each training batch. KL-Divergence is only defined for probability values, so the encoding space must be restricted to values between 0 and 1.

A popular variant of autoencoder is the variational autoencoder (VAE), which, instead of an encoding vector, produces a set of independent Gaussian distributions [35]. We looked at VAEs but have not produced any significant results.

*2.3. Data*

The datasets used in the following experiments—TREX13, PondEx09, PondEx10—were collected using the mechanism described in [36]. A tower containing sensors and electronics moved along a 20 m rail system. The electronics included 10 cm receiver, at 5 cm/s to allow for Synthetic Aperture Sonar processing of acoustic backscattering measurements. The tower transmitted a 6 ms pulse sliding from 1 to 30 kHz twice per second as it moved along the rail. The beam width was greater than 40°. The receiver had 6 vertically aligned apertures used to determine the return angle of the backscattering.

The data collection occurred at distances of 5, 10, 15, 20, 25, 30, 35 and 40 meters from the center of the rail for a variety of targets at a minimum of nine orientations for each target, −80° to 80° in 20° increments. A few targets have data for the remaining 100° to −100° orientations as well.

The TREX13 dataset [16] has data for 27 diverse targets with a total of 810 time/cross range measurements. The PondEx datasets [37] total 117 time/cross range measurements for 13 targets.

The measurements are 2D arrays of floating point values, with a time axis and a position axis associated with measurements along the 20 m rail. When visualized, the backscattering response takes a shape similar to that of a smile, as shown in Figure 1, so the data is called smile data. The smile data has very large array size, but the smiles generally fit within a 1024 × 1024 element bounding box, so it is cropped to that size. 1024 is a power of 2, allowing up to 10 decimations in convolutional layers. The cropped data has its Fourier transform calculated along the time axis to produce raw frequency data.

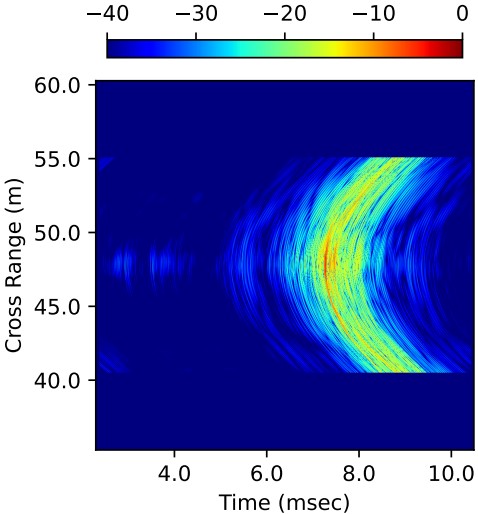

**Figure 1.** A "smile" image of a 155 mm Howitzer with collar (Target 9) from the TREX13 dataset, shown in decibels. This measurement was taken at 20 m and with a 0° inclination

The network implementations only handle real-valued data, so the data excluded the phase information, leaving the magnitude only. The magnitude of the frequency domain of a real signal is always symmetrical, so the training data only used half of the frequency data, leaving 512 (frequency) by 1024 (position) sized data arrays.

Visualization of magnitude frequency data is typically done in decibel form, so we also used this representation of the data as a training set, but the results were not useful.

### 2.4. Networks

We used the Python libraries Tensorflow [38] and Keras [39] to implement and train our networks through standard back propagation [40], using the Keras default for many metavariables. Further refinement of the metavariables is for future works.

We chose the Parametric ReLU (PReLU) [41] nonlinearity as our activation between convolutional layers as opposed to the more typical Rectified Linear Unit (ReLU) and Leaky ReLU [42]. PReLU is similar to Leaky ReLU, but the slope of the response to negative values is a trainable parameter–thus any activation can learn any range of activations between a typical ReLU to an identity, and beyond in both directions, as the problem needs.

The Glorot Uniform [43] weight initializer and the standard Adam optimizer [44], both with default parameters excluding learning rate, set and refined the learnable values weights.

The networks themselves are convolutional autoencoders acting upon $1024 \times 512$ input images, wherein an image in this application domain is described in Section 2.3. Our encoders have seven convolutional layers that each have $4 \times 4$ kernels, a stride factor of 2, and a batch normalization before a PReLU as the nonlinear activation.

The first layer reduces the $1024 \times 512$ input to a single channel $512 \times 256$ intermediate image, and subsequent layers each double the number of channels from their input to output.

After the seven convolutional layers, the 64 channels of $8 \times 4$ images are unwrapped into a single 2048 element vector. A neuron layer reduces this vector to a vector of the desired latent dimension. In the case of the basic autoencoder, there is no no-linearity applied to this layer, but for the sparse autoencoder a sigmoid function is applied to constrain the values between 0 and 1.

The decoder is close to an inverse network structure, with batch normalization followed by a PReLU functioning as a unit. It starts with a neuron layer that has batch normalization and a PReLU after it. The output of the neuron layer is reshaped into 64 channels of $8 \times 4$ images and fed into a sequence of transposed convolutional layers. The transposed convolutional layers have $4 \times 4$ kernels and a stride of 2 like in the encoder, and also have batch normalization followed by PReLU activations. The layers halve the number of channels from their input to output, except for the final layer which maintains a single channel. This final layer does not have batch normalization or PReLU applied to its output. The full network can be seen in Figure 2.

The integer ratio between the stride and kernel size is meant to reduce checkerboard noise patterns in the network output, as explored in [45]. Batch normalization is used increase the allowable learning rate magnitudes for training.

### 2.5. Training Process

The general training process is as follows:

1. Preprocess the data as described in Section 2.3.
2. Initialize a new autoencoder with random weights and the correct parameters.
3. Begin training the autoencoder on the full data set for 100 epochs (unless otherwise specified), using MSE loss.
4. Ensure that the loss value is beneath a threshold after one third of the training epochs have completed. Cancel the training and return to Step 2 if it is not.
5. Save the final weights of the trained network.
6. Repeat from Step 2 until ten sets of weights are saved.

When performing cross-validation, the process is slightly different. Rather than train on the full data set, five folds of the data set are isolated and the process from Step 2 and onward is performed on the each of the five complementary sets to the excluded folds. So, for each of the folds, the data not in the fold is used to train a network ten times.

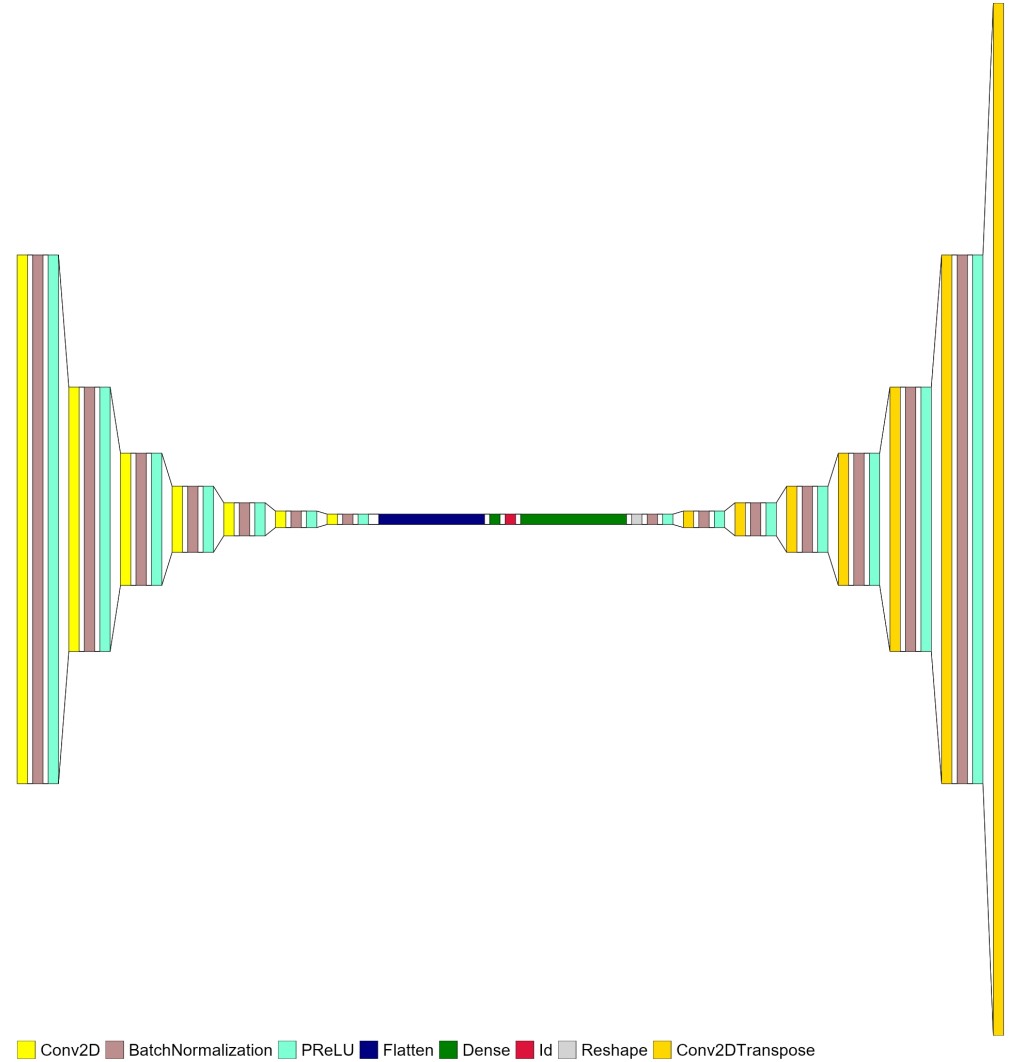

**Figure 2.** The structure of the convolutional autoencoder used in this paper. The encodings are output from the Identity (Id) block. After replacing the identity with a sigmoid activation, a sparsity constraint can be imposed on the network, resulting in a sparse convolutional autoencoder.

### 2.6. Evaluation

To obtain an encoding, an input is put through just the encoder half of the network. The decoder does not provide any use in this case.

Reconstruction accuracy across a reserved fold in cross-validation is estimated by calculating MSE between each data element in the fold and the reconstruction the network produces for it. Analysis is performed on the reconstruction accuracies across the whole fold.

## 3. Results

### 3.1. Learning Rate Determination

The optimal learning rate of our network and optimizer combination was unknown, so we tested a set of learning rates with various latent space dimensions. The learning rate tested were the set $\{1.0, 0.1, \ldots, 10^{-6}\}$ was and the latent space dimensions were

$\{2, 4, 6, \ldots, 16\}$. A comparison of the results of training five networks for 50 epochs each with each combination of the sets facilitated a choice.

As seen in Figure 3, the rate 0.001 shows the earliest descents of the loss, so we used it for the rest of the experiments. The difference between the basic autoencoder and the sparse autoencoder is small enough that the 0.001 learning rate is still adequate.

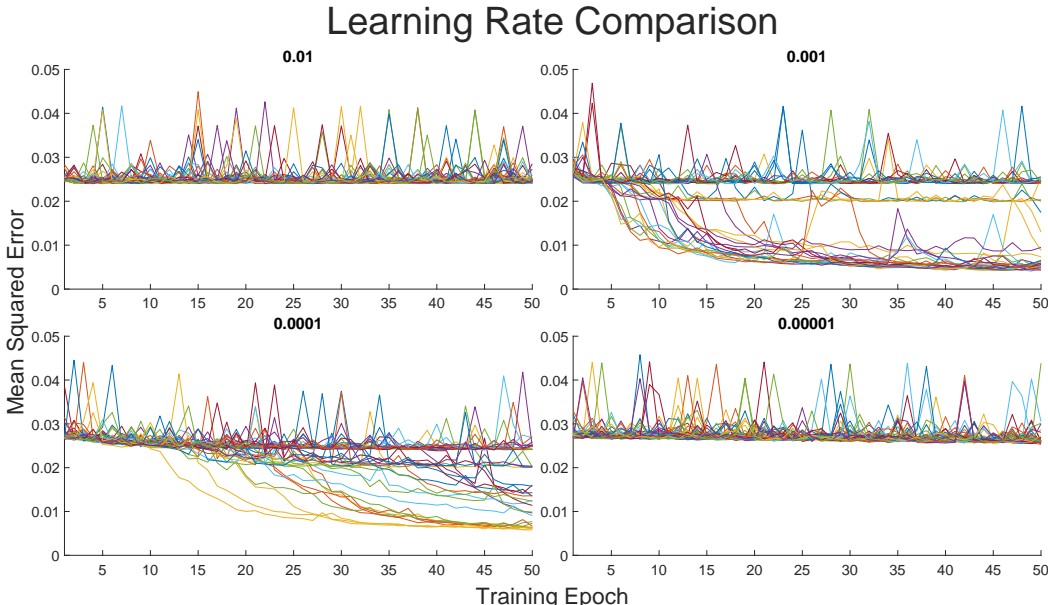

**Figure 3.** Plots of the loss trend lines for every model trained for some learning rates. The line colors serve only to differentiate different runs. The data for the other learning rates looks similar to the 0.01 and 0.00001 cases so it is excluded.

In the data for the chosen learning rate, there is a noticeable poor performing level of convergence at around the Mean Squared Error of 0.02. This likely indicates a set of sub-optimal local minima in the gradient equation for the encoding of the dataset, and a successful training of the network on the dataset must avoid this gradient trap. Trained encodings that end up at this or the higher convergence value are not useful, so any occurrence was rerun with different randomized initial weights.

*3.2. Latent Space Dimension Comparison*

We trained ten successful encodings for every even valued latent dimension from 2 to 30 for both the basic autoencoder and the sparse autoencoder. The MSEs of the dataset on the final learned encoding spaces and the best performing encoding space through the training process are shown in Figure 4. The same data is provided for the sparse encodings in Figure 5. The trend in MSE values as the latent encodings increase in dimension is similar looking to the reciprocal function for all four of the plots. By inspection, 16 appears to be a dimension choice that balances between high accuracy and low dimensionality.

For the sixteen dimension (16D) case, we give the reconstruction error for all combinations of measurement distance and angle in Table 1. Every measurement from the TREX13 data was put through the ten successfully trained 16D autoencoders and had a reconstruction error calculated. In comparison, the reconstruction errors for the 2D case are noticeably worse, as shown in Table 2.

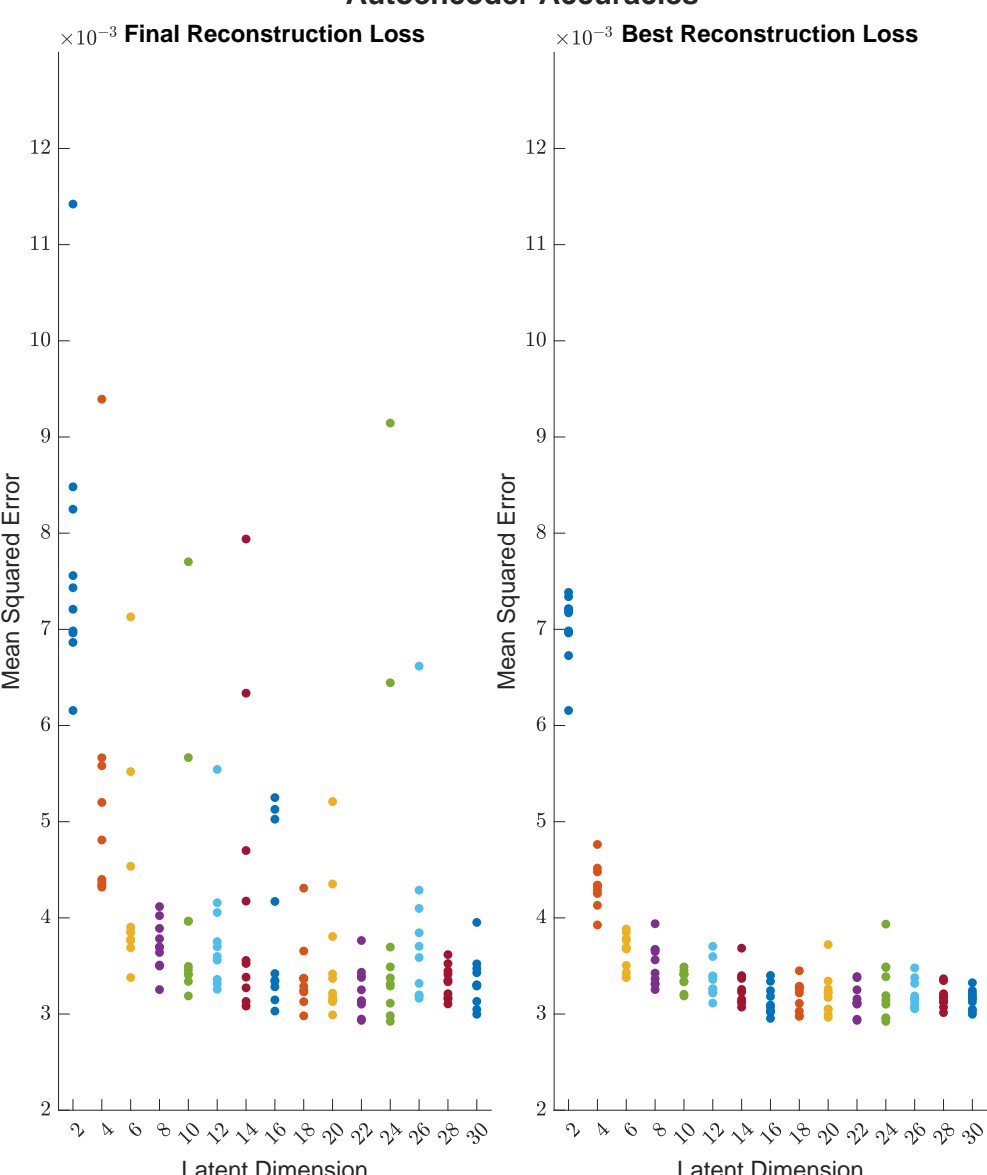

**Figure 4.** The Final and Minimum loss values across 10 successfully trained encodings for each tested latent space dimension. The colors differentiate adjacent columns.

*3.3. 2D Encoding Space*

16D encoding spaces are very hard to visualize, so for the sake of demonstration, 2D encodings will be evaluated. Figure 6 contains the full TREX13 dataset as encoded by the best performing of the ten 2D encodings from the previous section. The colors of the points match according to the measurement distances they are associated with in the dataset. Clearly there is a densely packed region of the encoding space where most of the dataset lies, with a few outlying points that tend to be in the 30 m–40 m measurement range.

The dense group is shown with more clarity in Figure 7. Here it is clear that the data measured at different distances forms bands that follow apparent contours from the 5m range to the 25 m range while the 30 m to 40 m data is not well separated, which is likely because the network cannot learn features that distinguish data measured at that far of a distance (low of a resolution). It is curious that the encodings at the other five distances appear in the correct order, and this was consistent across all of the learned 2D encoding spaces, see Figure 8.

**Figure 5.** The Final and Minimum loss values across 10 successfully trained sparse encodings for each tested latent space dimension. The colors differentiate adjacent columns.

Due to the nature of the data, it is very easy to distinguish measurement distance based on the size of the zero padding. It makes sense that the lowest dimension encoding is able to at minimum distinguish most of the distances. Additionally, different distances need features to be learned at different scales, so different parts of the network are activated, thus they have different encodings.

The outlying encodings are consistent between the learned encoding spaces. Figure 9 shows that the four encodings farthest from the mean are the same in the best and worst performing of the encoding spaces. The significance that all four are of target type 1, which is a DEU trainer, will not be explored here.

We also evaluated the sparse 2D encoding spaces; the best performing space is shown in Figure 10. Here we see that the sparsity constraint restricted the values of the encoding vectors to between 0.0 and 0.1. This forced the previous outlier points to stay closer to the mean value. Additionally, the contours of distance measurement groups are not as separated.

**Table 1.** The autoencoder's average normalized reconstruction errors with 16 latent dimensions. All values $\times 10^{-3}$.

|  | **5 m** | **10 m** | **15 m** | **20 m** | **25 m** | **30 m** | **35 m** | **40 m** |
|---|---|---|---|---|---|---|---|---|
| $-80°$ | 0.133 | 0.686 | 1.716 | 4.021 | 4.883 | 9.283 | 22.009 | 22.009 |
| $-60°$ | 0.149 | 0.698 | 1.678 | 3.131 | 3.676 | 4.449 | 5.891 | 5.891 |
| $-40°$ | 0.129 | 0.515 | 1.928 | 2.502 | 3.800 | 4.153 | 4.998 | 4.998 |
| $-20°$ | 0.200 | 0.775 | 3.169 | 3.520 | 6.335 | 7.475 | 10.532 | 10.532 |
| $0°$ | 0.278 | 1.157 | 3.441 | 4.976 | 8.483 | 9.167 | 14.659 | 14.659 |
| $20°$ | 0.278 | 1.083 | 2.414 | 4.095 | 5.832 | 7.842 | 8.907 | 8.907 |
| $40°$ | 0.159 | 0.674 | 1.871 | 2.724 | 4.372 | 4.622 | 5.072 | 5.072 |
| $60°$ | 0.172 | 0.617 | 1.278 | 2.820 | 4.183 | 4.084 | 5.949 | 5.949 |
| $80°$ | 0.202 | 0.592 | 1.497 | 4.268 | 4.900 | 6.701 | 11.929 | 11.929 |
| $100°$ | – | – | 0.175 | – | – | – | 6.245 | 6.245 |
| $120°$ | – | – | 0.428 | – | – | – | 6.168 | 6.168 |
| $140°$ | – | – | 0.576 | – | – | – | 4.361 | 4.361 |
| $160°$ | – | – | 0.902 | – | – | – | 7.110 | 7.110 |
| $180°$ | – | – | 1.475 | – | – | – | 16.876 | 16.876 |
| $200°$ | – | – | 0.895 | – | – | – | 6.671 | 6.671 |
| $220°$ | – | – | 0.577 | – | – | – | 4.662 | 4.662 |
| $240°$ | – | – | 0.477 | – | – | – | 10.769 | 10.769 |
| $260°$ | – | – | 0.772 | – | – | – | 22.848 | 22.848 |

**Table 2.** The autoencoder's average normalized reconstruction errors with 2 latent dimensions. All values $\times 10^{-3}$.

|  | **5 m** | **10 m** | **15 m** | **20 m** | **25 m** | **30 m** | **35 m** | **40 m** |
|---|---|---|---|---|---|---|---|---|
| $-80°$ | 0.190 | 1.283 | 2.948 | 8.062 | 10.700 | 20.616 | 29.097 | 29.097 |
| $-60°$ | 0.204 | 0.864 | 1.956 | 4.725 | 5.783 | 7.227 | 8.019 | 8.019 |
| $-40°$ | 0.174 | 0.653 | 2.280 | 3.859 | 5.277 | 7.539 | 6.473 | 6.473 |
| $-20°$ | 0.264 | 1.106 | 6.529 | 6.471 | 15.699 | 16.945 | 30.716 | 30.716 |
| $0°$ | 0.463 | 1.984 | 6.934 | 9.838 | 15.538 | 24.862 | 34.056 | 34.056 |
| $20°$ | 0.399 | 1.833 | 3.815 | 7.150 | 12.524 | 18.244 | 23.496 | 23.496 |
| $40°$ | 0.214 | 0.825 | 2.468 | 4.635 | 7.978 | 8.916 | 7.018 | 7.018 |
| $60°$ | 0.257 | 0.787 | 1.537 | 3.891 | 5.698 | 5.559 | 12.711 | 12.711 |
| $80°$ | 0.312 | 1.210 | 2.070 | 8.857 | 13.441 | 17.535 | 25.996 | 25.996 |
| $100°$ | – | – | 0.269 | – | – | – | 10.247 | 10.247 |
| $120°$ | – | – | 0.551 | – | – | – | 7.455 | 7.455 |
| $140°$ | – | – | 0.667 | – | – | – | 5.502 | 5.502 |
| $160°$ | – | – | 1.532 | – | – | – | 9.789 | 9.789 |
| $180°$ | – | – | 2.448 | – | – | – | 54.332 | 54.332 |
| $200°$ | – | – | 1.668 | – | – | – | 8.022 | 8.022 |
| $220°$ | – | – | 0.754 | – | – | – | 5.612 | 5.612 |
| $240°$ | – | – | 0.609 | – | – | – | 16.081 | 16.081 |
| $260°$ | – | – | 0.956 | – | – | – | 29.829 | 29.829 |

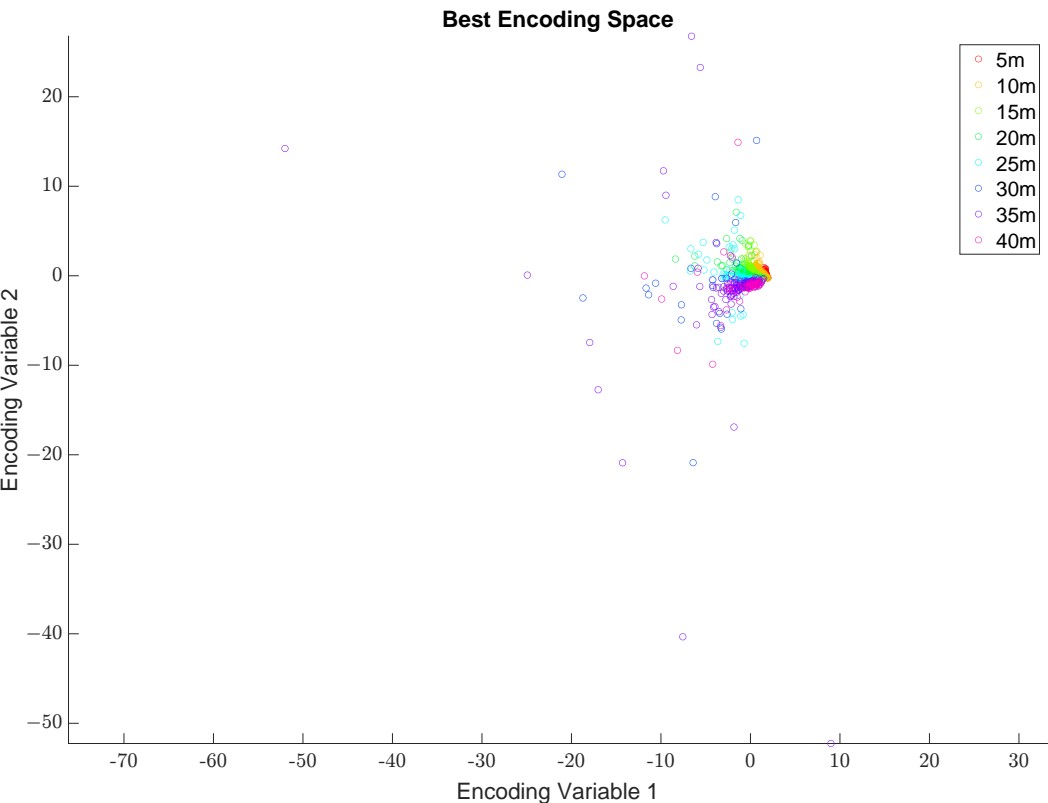

**Figure 6.** The best performing 2D encoding space, showing the encodings of the input data. These are color coded by the measurement distance of the images.

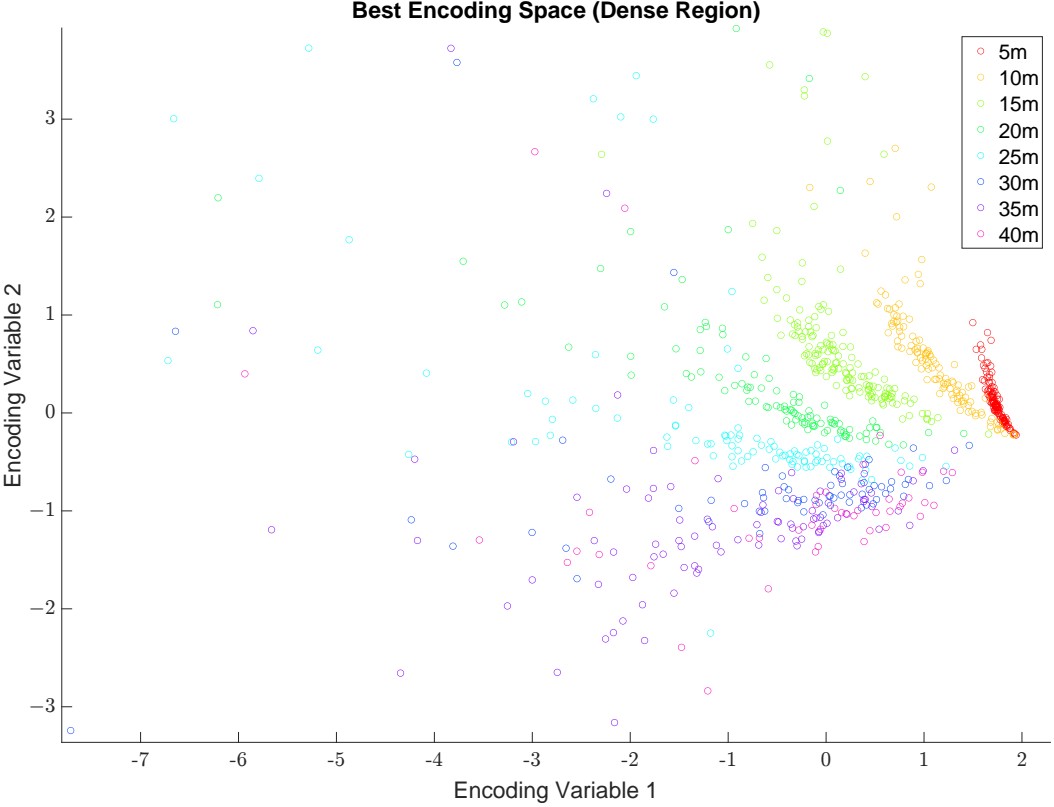

**Figure 7.** A close up of the tight group of encodings in Figure 6.

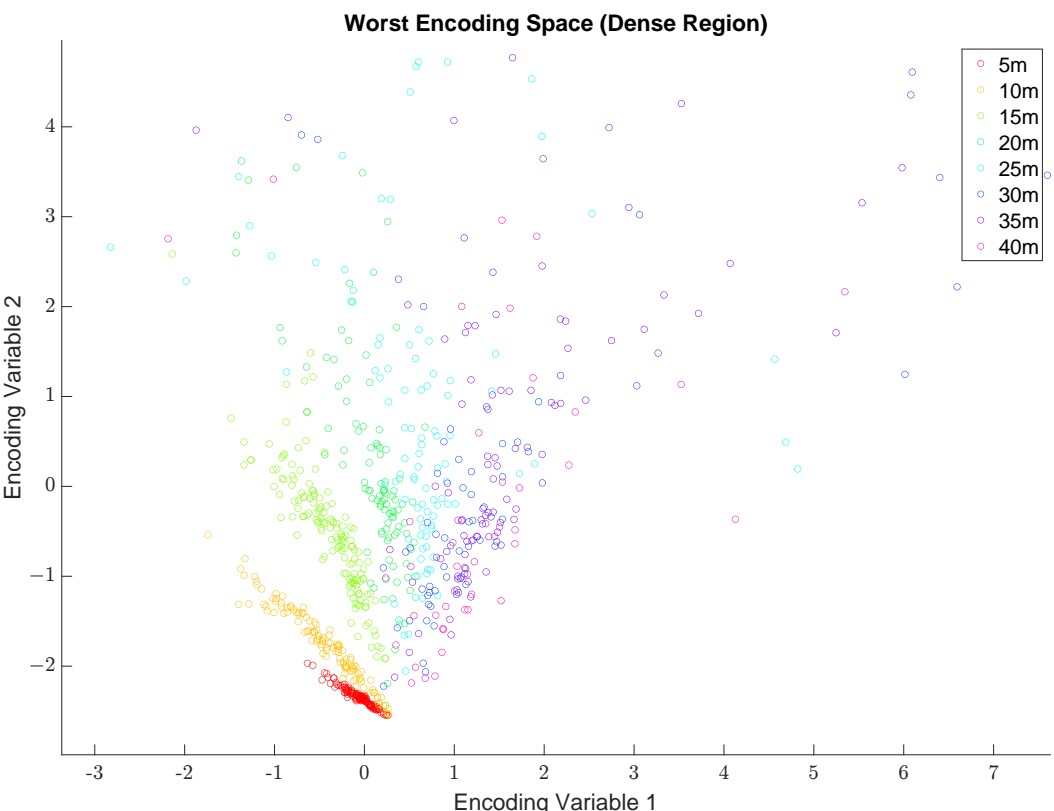

**Figure 8.** The dense part of the worst performing 2D encoding space.

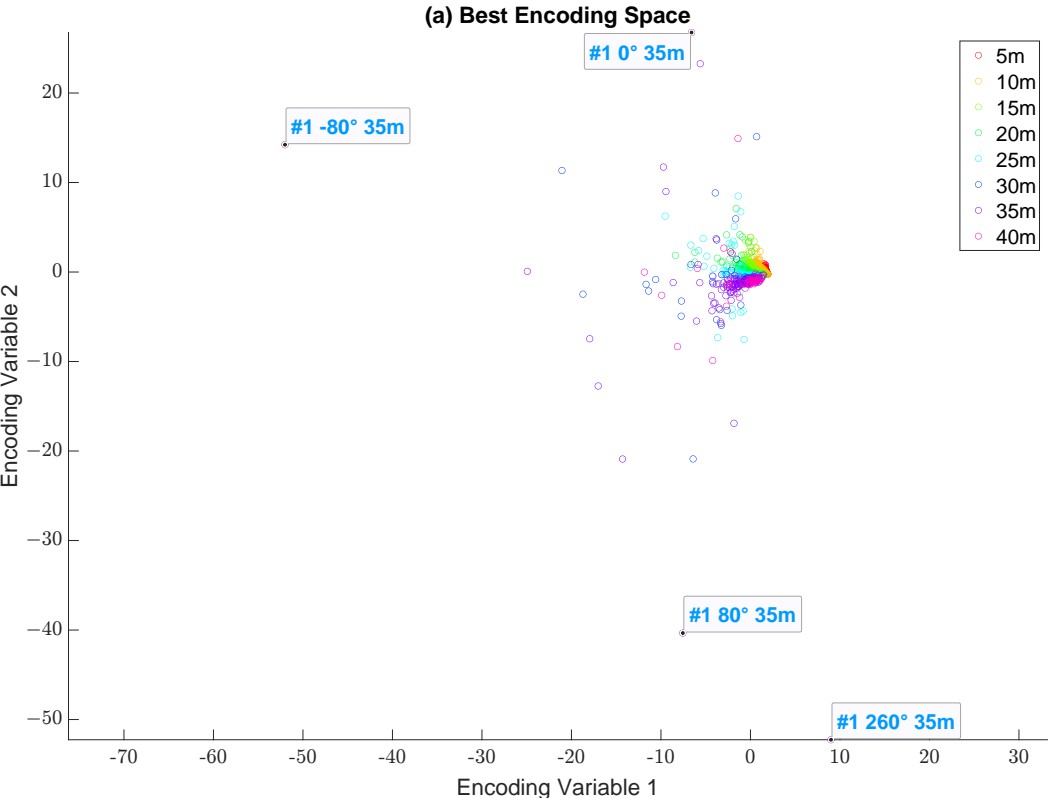

**Figure 9.** *Cont.*

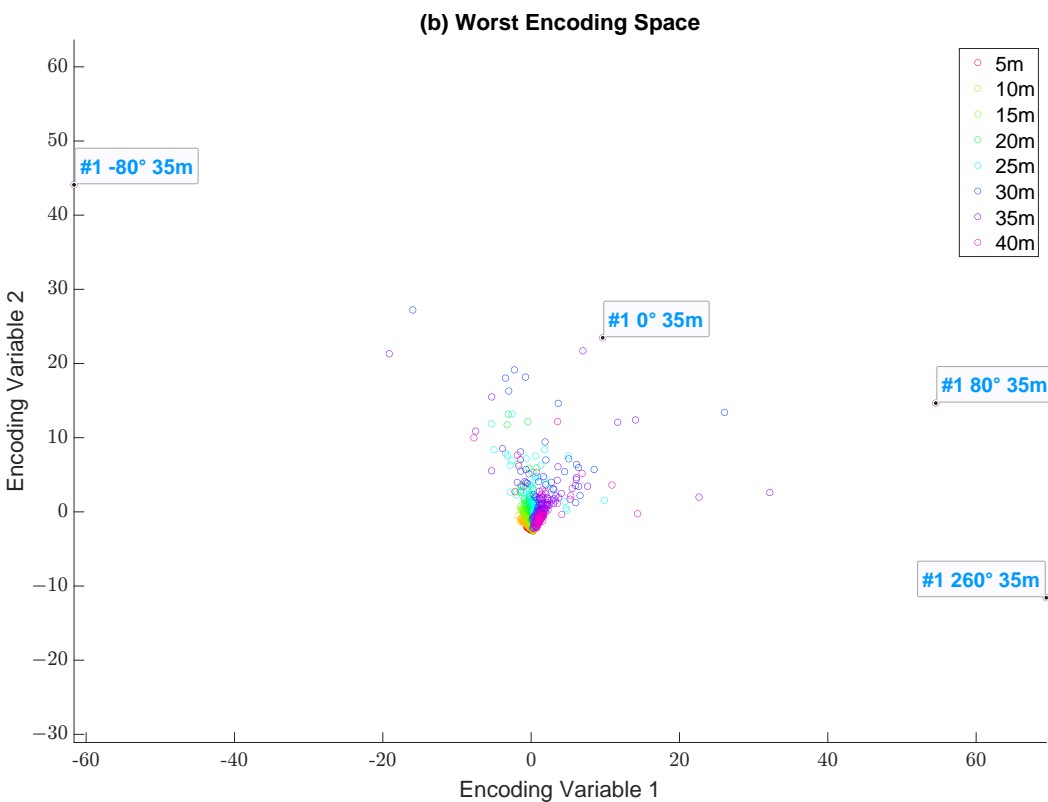

**Figure 9.** A comparison of outlier encodings between the best (**a**) and worst (**b**) performing encoding spaces. The labels indicate target number, measurement angle, and measurement distance.

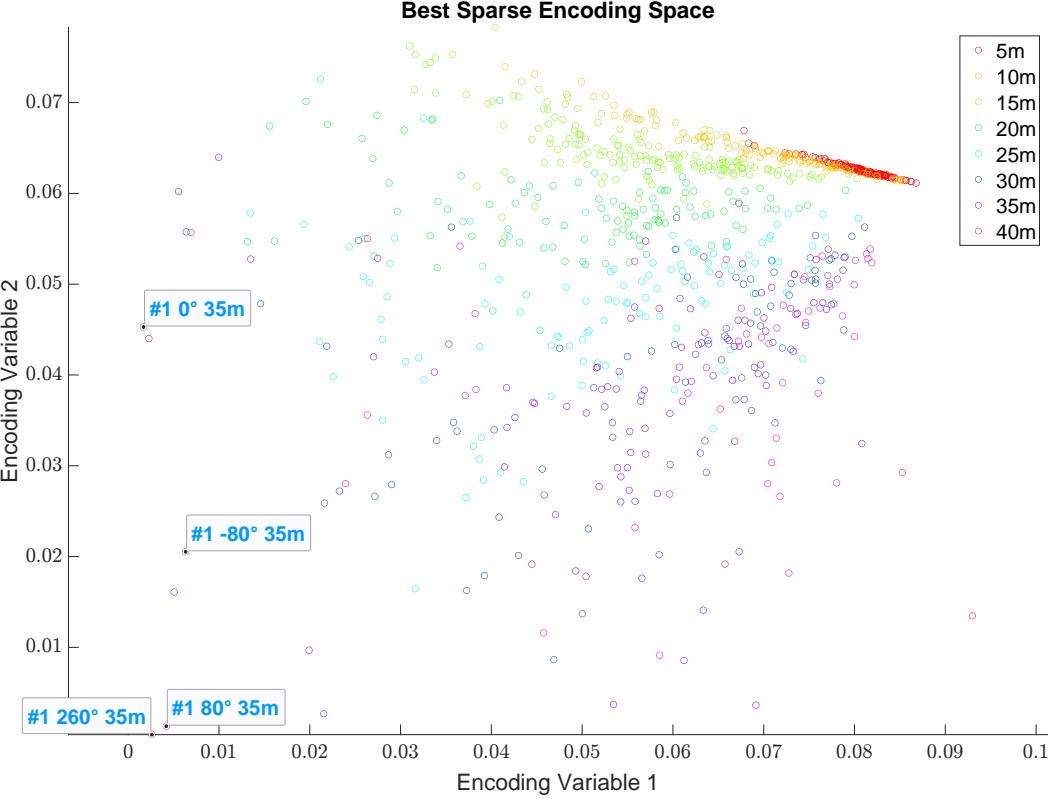

**Figure 10.** The best performing sparse 2D encoding space. The encoding points of the same four outlier inputs labeled in previous figures are also labeled here for comparison.

### 3.4. Performance on Unseen Data

With a choice of 16 dimensions in the encoding space, we ran 5-fold cross-validation on the TREX13 dataset. The results of this are shown in Figure 11 for the autoencoder and sparse autoencoder.

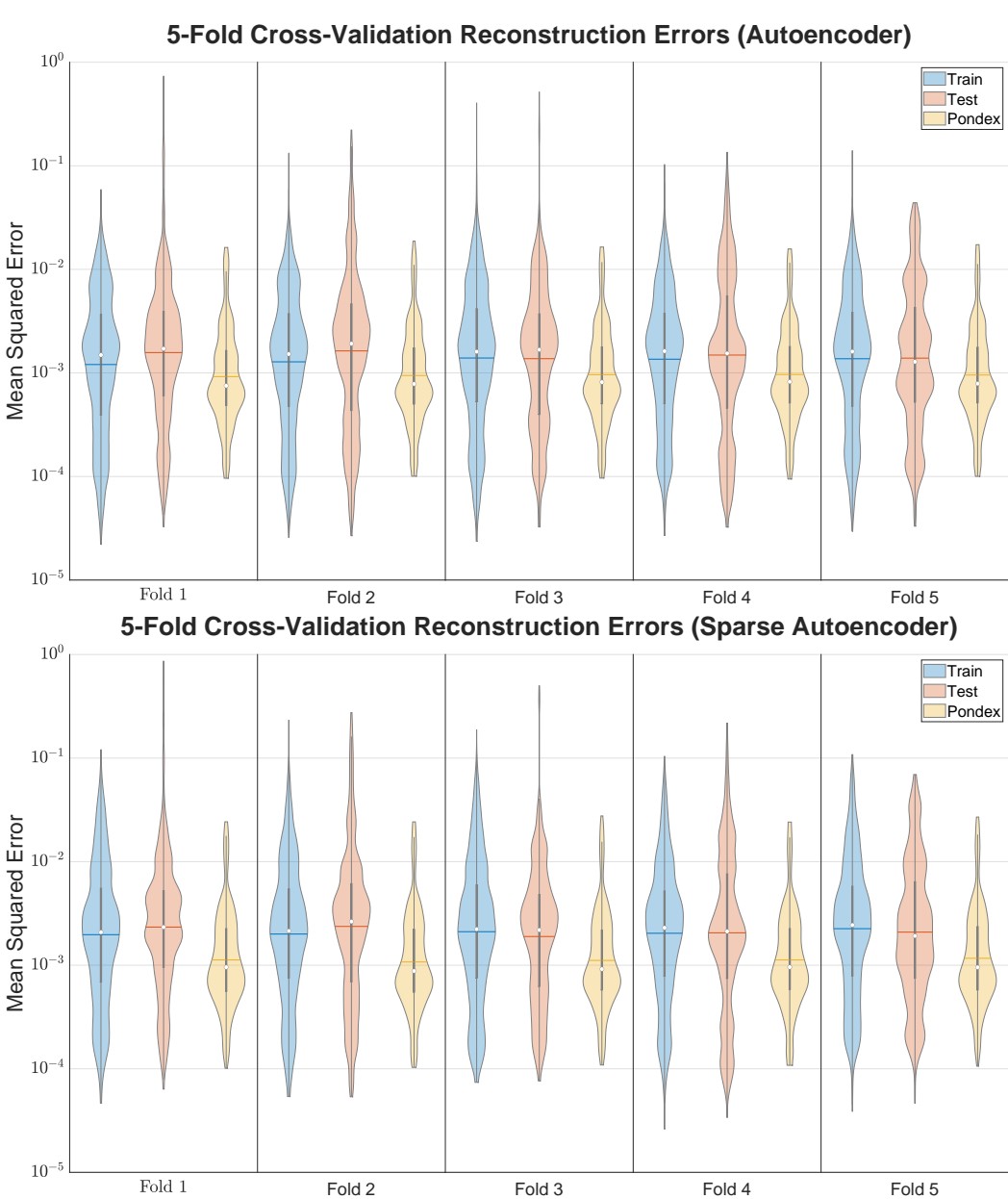

**Figure 11.** 5-fold cross validation across the TREX13 dataset, also performance on PondEx data.

Across both autoencoders, there were five successfully trained encodings for each of the five folds. The mean squared reconstruction error for each individual magnitude Fourier transform in the training sets and test sets was obtained for each of the 25 total encoding spaces.

Interestingly, the set of these error values fits a logarithmic scale despite the networks being trained on non-decibel data. In Figure 11, the base-10 logarithms of the reconstruction errors are visualized with violin plots, with the y axes corrected to show the values on a logarithmic scale. Violin plots are similar to box plots with the benefit of fitting multi-modal data far better. Vertically they show an estimated density, in this case the density estimation kernel is 10% of the dataset. The violin plots were generated with an external Matlab library [46]. In these violins the data presented by a box plot is also shown; the white circles

represent the medians and the gray vertical lines show the inner quartile range. The dark colored horizontal lines indicate the mean values.

The experiments with the autoencoders and sparse autoencoders used the same training and testing folds, which explains why the shapes of the violins associated with the folds look similar between the two networks. Looking at just the means and medians of the violins, in some of the folds the test set outperforms the training set. This shows that the learned encoding spaces generalize well to unseen data in the same dataset.

The logarithm of the reconstruction performance of the PondEx dataset is also shown in this figure with yellow violins. The 25 autoencoders and sparse autoencoders encoded the entire previously untouched PondEx dataset, producing similar looking violins regardless of fold. Notably, the PondEx data appears to achieve better reconstruction than the data that the autoencoders were trained on. The PondEx data was obtained in a controlled pool environment, whereas TREX13 was recorded in the ocean.

The PondEx data only contains data collected at either 5 and 10 m, so the distances that produced the largest error values in Table 1 are not represented. This suggests a skew due to selection bias. In Tables 3 and 4, we can see that while the PondEx data performs fairly well at 5 m it does significantly worse than the training data at the 10 m range.

**Table 3.** Average error values for the PondEx datasets across all cross-validation folds (basic convolutional autoencoder). All values $10^{-3}$.

|       | $-80°$ | $-60°$ | $-40°$ | $-20°$ | $0°$  | $20°$ | $40°$ | $60°$ | $80°$ |
|-------|--------|--------|--------|--------|-------|-------|-------|-------|-------|
| 5 m   | 0.202  | 0.172  | 0.254  | 0.412  | 0.307 | 0.275 | 0.129 | 0.164 | 0.189 |
| 10 m  | 1.537  | 1.034  | 1.205  | 3.662  | 2.588 | 3.383 | 1.376 | 1.459 | 2.305 |

**Table 4.** The low measurement distance data from Table 1. All values $\times 10^{-3}$.

|       | $-80°$ | $-60°$ | $-40°$ | $-20°$ | $0°$  | $20°$ | $40°$ | $60°$ | $80°$ |
|-------|--------|--------|--------|--------|-------|-------|-------|-------|-------|
| 5 m   | 0.133  | 0.149  | 0.129  | 0.200  | 0.278 | 0.278 | 0.159 | 0.172 | 0.202 |
| 10 m  | 0.686  | 0.698  | 0.515  | 0.775  | 1.157 | 1.083 | 0.674 | 0.617 | 0.592 |

## 4. Conclusions

The goal of ongoing work is a disentanglement of the littoral channel noise from the true sonar response. We have demonstrated significant progress towards encoding underwater backscattering data. The precursory results show promise by demonstrating clustering correlated to measurement distance and the ability to generalize to unseen data. The optimal dimensionality of the encoding space has been estimated to be around sixteen dimensions, Section 3.2, and the efficacy of this latent vector rank was validated through cross validation in Section 3.4. While the basic convolutional autoencoders show promise, the sparse autoencoders do not provide tangible benefits yet in training efficiency or accuracy, but a constrained encoding space may yet prove useful in the future.

Future steps may involve developing alternatives to convolutional neurons or adding additional preprocessing steps to the data before training on it. Complex neural networks are one path that can be explored; this would include the phase information from the frequency domain data that is currently excluded at the cost of increasing the amount of operations needed per layer during both the forward and back propagation steps of the training.

The Keras default values were used in many places. In addition to fundamental changes like neuron type, variations on the metavariables may improve reconstruction accuracy and reduce training time. Candidates for this include: adjusting the network to have double the kernel count in the encoders so that the network is more symmetrical, using the He method [41] for weight initialization, changing which activation functions to use–PReLU may add unneeded complexity over a ReLU, varying kernel shape and size as

relevant features may not be square, and fluctuating the number of convolutional layers in both the encoder and decoder.

**Author Contributions:** Conceptualization, A.S.G.; Methodology, T.J.L.; Software, T.J.L.; Validation, T.J.L.; Formal analysis, T.J.L.; Investigation, T.J.L.; Resources, A.S.G.; Writing—original draft, T.J.L. and M.B.; Writing—review & editing, A.S.G. and M.B.; Visualization, T.J.L.; Supervision, A.S.G.; Project administration, A.S.G.; Funding acquisition, A.S.G. All authors have read and agreed to the published version of the manuscript.

**Funding:** This research was funded by The Office of Naval Research grant number N00174-20-1-0016.

**Institutional Review Board Statement:** Not applicable.

**Informed Consent Statement:** Not applicable.

**Data Availability Statement:** Not applicable.

**Conflicts of Interest:** The authors declare no conflict of interest.

## Abbreviations

The following abbreviations are used in this manuscript:

| | |
|---|---|
| PCA | Principal Component Analysis |
| ReLU | Rectified Linear Unit |
| PReLU | Parametric Rectified Linear Unit |
| MSE | Mean Squared Error |
| ATR | Automated Target Recognition |
| GAN | Generative Adversarial Network |
| VAE | Variational Autoencoder |
| UXO | Unexploded Ordinance |

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
