# Peer review of "Convolutional Autoencoding of Small Targets in the Littoral Sonar Acoustic Backscattering Domain"

_jmse, doi:10.3390/jmse11010021_

Round 1

Reviewer 1 Report

In principle the presented results could be of interest for JMSE readers. 

In general, the manuscript is original, significant and well-written. 

I would support its publication after some minor improvements of presented figures.

1) Figure 1 has large scale and uses very large fonts.

2) Figure 2 is qualitative enough

3) Figure 3 uses very small fonts.

4) Figure 4 is so large in compare with plots on Figure 3.

5) Figure 4 is not different from upper right plot on Figure 3.

I don't why there is the same plot in different figures.

6) Figure 5 - 11  have style "no box", but Figure 3 and Figure 4 have style "box".  I suppose that all figures have to be same style.

7) Figure 7 - 12  have large scale and uses very small fonts.

It seems to me the paper figures have to be improved.

I suppose that section "Discussion" should be renamed  to "Conclusion" and it's text should be definitely improved.

There is no section paper - "Author Contributions".

Reviewer 2 Report

In this paper, the authors discussed convolutional autoencoding of small targets in the littoral sonar acoustic backscattering domain. In general, the topic in this paper is important, and the authors presented many useful results. However, the method in this paper is not well presented. At this point, the reviewer has two queries.

1. In introduction, the work related to authors’ method should be comprehensively described. The advantages and disadvantages should be clearly clarified.

2. The method in this paper should be re-described, as the reviewer cannot find the processing steps of their method. A block diagram of their method should be presented.
